# The cost of electrifying all households in 40 Sub-Saharan African countries by 2030

**Florian Egli** [1,2] ✉, **Churchill Agutu** [1,3] ✉, **Bjarne Steffen** [4,5,6] ✉ & **Tobias S. Schmidt** [1,5,7,8] ✉

Electrifying sub-Saharan Africa (SSA) requires major investments and policy intervention. Existing analyses focus on the levelized cost of electricity at aggregate levels, leaving the feasibility and affordability of reaching Sustainable Development Goal #7 – access to affordable, reliable, sustainable and modern energy for all – by country unclear. Here, we use the electrification model OnSSET to estimate granular and spatially explicit levelized costs of electricity and costs per person per day (pp/d) for 40 countries in SSA. We find that solar-powered mini-grids and standalone systems drastically lower the cost of electrifying remote and high-cost areas, particularly for lower tiers of electrification. On average, least-cost electrification in SSA at Tier 3 (ca. 365 kWh/household/year), can be provided at 14c USD/kWh or 7c USD pp/d. These results are sensitive to demand assumptions, for example, misguided electrification planning or oversizing due to overestimated demand can lead to substantial cost increases. Our results highlight large variances within countries, which we propose to visualise using electrification cost curves by country. Policymakers should consider such cost curves and use a tailored approach by country and region to reach SDG7 in SSA.

It is estimated that 752 million people lacked access to electricity in 2020, out of which 581 million (77%) live in sub-Saharan Africa (SSA)[1]. Providing access to "affordable, reliable, sustainable and modern energy for all" is a key development goal as stipulated in United Nations' Sustainable Development Goal (SDG) 7, and a precondition for achieving the majority of SDG targets[2]. Target 1 of SDG7 focuses on "ensur[ing] universal access to affordable, reliable and modern energy services [by 2030]" and for SSA, electrification is particularly important due to the prevailing challenges in achieving many development outcomes[3]. Moreover, electricity demand is expected to grow massively on the African continent in the coming decades[4]. Therefore the emission intensity of the continent's electrification path will have implications for low-carbon development and eventually climate change too[5–7], which can hamper progress on virtually all SDGs[8].

Progress on electrification in SSA has been steady in the past, but in 2020 due to the Covid-19 pandemic and continuing population growth, the number of people without access to electricity in SSA increased for the first time since 2013[4,9]. Government budgets remain constrained due to the economic downturn and unsustainable debt levels hinder public investment in many developing countries[10], which puts reaching SDG7 further at risk[11,12]. At the same time, components for off-grid electrification technologies – particularly solar photovoltaic (PV) modules and batteries – have experienced massive cost reductions in recent years[13]. Helped by simultaneous business model innovation in off-grid electrification delivery[14–16], total costs for low-

[1]Energy and Technology Policy Group, ETH Zurich, Zurich, Switzerland. [2]IIPP Institute for Innovation and Public Purpose, UCL, London, UK. [3]Kigali Collaborative Research Center, Kigali, Rwanda. [4]Climate Finance and Policy Group, ETH Zurich, Zurich, Switzerland. [5]Institute for Science, Technology and Policy, ETH Zurich, Zurich, Switzerland. [6]Center for Energy and Environmental Policy Research, Massachusetts Institute of Technology, Cambridge, MA, USA. [7]Andlinger Center for Energy and the Environment, Princeton University, Princeton, NJ, USA. [8]Center for Policy Research on Energy and the Environment, Princeton University, Princeton, NJ, USA. ✉e-mail: florian.egli@gess.ethz.ch; churchill.agutu@gess.ethz.ch; bjarne.steffen@gess.ethz.ch; tobiasschmidt@ethz.ch

carbon off-grid electrification have decreased dramatically, opening up new opportunities for electrification beyond the traditional route of grid extension[16].

These new electrification approaches could allow reaching SDG7 and consequently a variety of SDGs at a lower cost. However, to date it remains unclear what the cost of electrification to the consumer is at a geographically granular level. This information is crucial because previous research has shown that least-cost electrification choices are highly context dependent[17–19] and yet policymakers do not know whether local consumers will be able to bear the cost of the provided electrification[20]. Without this information, policymakers risk building out infrastructure that remains inaccessible to the population because of a lack of ability to pay for it – often referred to as the affordability gap[21]. This jeopardises electrification targets and is highly troublesome for low-income countries with constrained budgets.

Here, we use the open-source electrification tool OnSSET (see methods), to provide granular and spatially explicit cost estimates for electrification in SSA. We produce SSA-wide and country-specific electrification cost curves, which help identify areas where the ability to pay may not be sufficient given the cost of the cheapest available electrification option. The model calculates the least-cost electrification option between grid extension (GE), hydro or solar PV mini-grid (MG), and solar PV standalone system (SAS) using a population raster layer at a resolution of 100 m × 100 m to achieve 100% electrification by 2030 as stipulated in SDG7. It considers population growth, population density, realistic financing conditions[17] and further techno-economic variables, such as technology cost[17] (see methods). We consider different levels of energy access, using the World Bank's Multi-Tier Framework for Energy Access (MTF)[22] which provides a classification of typical annual household electricity consumption based on reliability and quality of electricity services (see Supplementary Table S1).

First, we show that the availability of low-cost, low-carbon and off-grid electrification options (MG and SAS) can drastically lower the cost of electrifying remote and high-cost areas. Second, our results suggest that cost-optimal electrification mainly takes place via grid densification, grid extension and SAS unless households upgrade to Tier 4 consumption, which is unlikely for many rural areas in SSA within the decade (note that Tier 4 requires at least 3.4 kWh per household per day, equalling roughly the average consumption in Indonesia or Tunisia)[22]. Third, we demonstrate that costs per person per day (pp/d) remain below USD 5c for all of SSA for basic access (Tier 2). When upgrading to Tier 3, costs remain relatively low at USD 16c pp/d maximum. Finally, we show substantial cost variation between and within country, with average Tier 3 electrification costs ranging from USD 3c pp/d in Gabon to USD 16c pp/d in Eswatini. These results suggest that reaching SDG7 does not lead to excessive access costs if policymakers make use of available off-grid technologies where these can reduce costs substantially.

## Results
### Electrification cost curves
Figure 1 illustrates the cost savings from off-grid technologies. To this end, it depicts the electrification cost curves if off-grid options were not available (grid only), if only grid and MG were available, and if grid, MG, and SAS were available for electrification. The levelized cost of electricity (LCOE) to electrify with grid only increases dramatically at around 500 million people. Adding MG, shifts this bending point to around 750 million, whereas further adding SAS results in LCOE of less than 0.50 USD/kWh for the whole unelectrified population. Put differently, the availability of low-cost SAS dramatically reduces the variation in electrification cost across SSA because it is a "cost leveller" for rural areas in countries with adverse investment environments and low population densities. Using all three options, we estimate that reaching 100% Tier 3 electrification by 2030 requires a total investment of USD

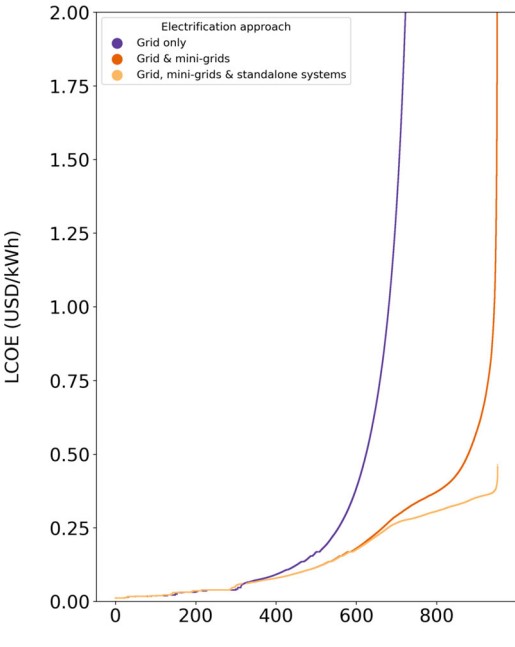

**Fig. 1 | Electrification cost curve for different sets of technologies.** Population plotted on the x-axis refers to the population to be electrified by 2030 incorporating population growth and current electrification levels. Results are shown for Tier 3. Note that due to population growth in currently unelectrified areas, the number of connected people (x-axis) is higher than the currently unelectrified population (ca. 580 million).

203bn. The availability of low-cost SAS reduces the investment to reach 100% electrification by USD 46 bn or 18% (comparing the investment to grid only is impossible because grid extension costs to remote areas would be exorbitant). While the reduction in investment is substantial, the reduction in the LCOE is massive for people living in rural, sparsely populated areas of countries with adverse investment environments. Given political economy and budgetary constraints[23,24] to provide electrification to people living in such areas, SAS substantially increases the chances of meeting SDG7.

These costs are subject to uncertainties, shown in Fig. S5 in the Supplementary Information. Namely, SAS input parameters (CAPEX and cost of capital) have the highest impact on average LCOE because high-cost areas are electrified with SAS, making these costs disproportionally important for the overall cost of electrifying SSA. Average LCOE would only increase by 8% if the CAPEX of SAS were to be 20% higher than assumed (see Table S4 for cost inputs). In turn, average LCOE reductions of 20% are possible if CAPEX were 20% lower than anticipated and substantial cost savings are possible if the cost of capital can be lowered. Grid and transmission and distribution (T&D) infrastructure cost inputs have a relatively large impact too because a large share of the unelectrified population is projected to receive grid electrification (see Fig. 2). Finally, MG cost inputs matter proportionally less because fewer people are electrified with MG. Accordingly, the sensitivities are higher for higher-cost areas (i.e., towards the right of electrification cost curves in Fig. S6 in the Supplementary Information). This shows that as electrification efforts reach higher-cost (i.e., typically more remote) areas, the uncertainties about costs also increase. Overall, Fig. S6 shows that the results are very robust with electrification curves moving only slightly as input parameters change.

While above we focused on Tier 3, we next compare the electricity cost curves considering all three electrification approaches for three demand levels (Tier 2 to 4) as the literature has already shown large differences across tiers[18,25,26]. Figure 2 shows the LCOE (2a), the cost per person per day (2b) and the shares for each electrification option (2c)

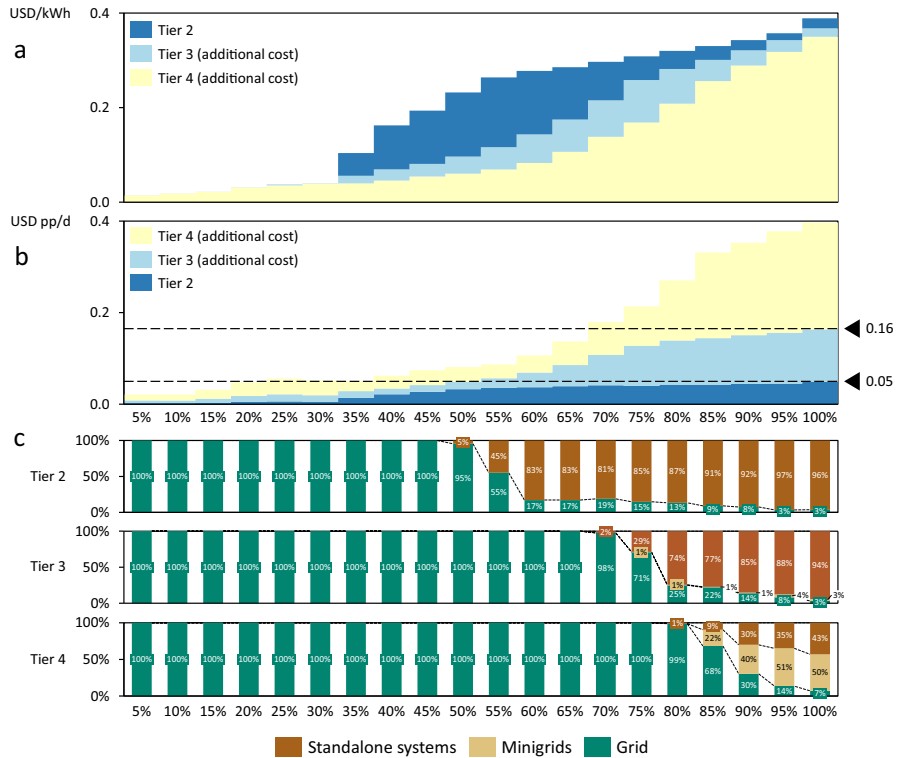

**Fig. 2 | Electrification cost curves and cost-optimal approaches. a** Electrification LCOE curve for SSA by tier. **b** Electrification cost per person per day curve for SSA by tier. Dashed lines show maximum costs for Tier 2 and Tier 3. **c** Cost-optimal electrification approach for SSA by tier. X-axis show ventiles (5% increments) of the population to be electrified by 2030 in SSA. Each ventile represents ca. 47.6 million people with a total to-be electrified population of 952 million people. Note that this number is larger than the currently unelectrified population because it includes population growth.

for each 5% increment of the unelectrified population in SSA. Generally, lower tier electrification results in higher LCOEs because the capital investment is high in relation to the electricity output over the lifetime of the asset. Put differently, the specific cost of higher consumption levels is lower due to economies of scale (see Fig. 2a). Maximum costs per person per day reach USD 5c pp/d for Tier 2, USD 16c pp/d for Tier 3 and USD 40c pp/d for Tier 4 (see Fig. 2b). Affordability remains a real challenge especially at higher tiers because 389 million people still live below the poverty line of USD 2.15 per day as defined by the World Bank[27]. These people are, to a large extent, likely to be living without access to electricity[28] with a low ability or willingness to pay for electricity access[20,29]. Total investment costs also increase substantially for higher tiers ($87bn for Tier 2, $203bn for Tier 3, and $408bn for Tier 4), and raising capital is still an issue in many SSA countries due to the difficult institutional environment[21,30]. Finally, the shares of the different electrification approaches vary considerably depending on the demand level (see Fig. 2c). SAS account for larger shares the lower the tier because the sizing of SAS allows for a better utilisation compared to grid electrification. MG shares are low except for high-cost areas in Tier 4, where high demand and the benefits of shared infrastructure costs outweigh the negative impact of higher financing costs (see methods). These demand Tier variations can represent the variance in small-scale productive use (e.g., micro-businesses operating from homes)[31], which would typically be situated between Tier 3 and Tier 4 (see examples in Supplementary Table S1). Beyond average demand levels, micro-businesses can also offer a benefit due to increased demand during the day, which has been shown for the case of MGs in Tanzania[32]. However, such an effect could also be expected for larger SAS and its quantification and generalisation across SSA is a task for future research.

While electricity demand can be underestimated if small-scale productive uses develop, empirically, demand is often below the initial projections of household electrification programmes in ex-post evaluations[33–35]. If demand is lower than anticipated, electrification may turn out more expensive because a suboptimal electrification approach is chosen (error 1: misguided planning) and/or because the chosen electrification approach is oversized (error 2: oversizing). We approximate these errors as follows. For error 1, we showcase a situation where the electrification approach is chosen optimally to achieve Tier 3 (or Tier 4) electrification, but effective demand is only Tier 2. We modelled this by contrasting the LCOE for Tier 3 (or Tier 4) electrification in each spatial cluster with the LCOE of the same, and potentially suboptimal, approach in Tier 2 (see Supplementary Table S5). We show that Tier 3 electrification leads to a 32% increase in the average LCOE across SSA if demand is only Tier 2 (86% for Tier 4). For error 2, we show LCOEs for a representative MG and SAS cluster set at the median Tier 3 LCOE for each electrification approach (see Supplementary Table S6). We contrast this with a case where demand amounts to 50% of the estimated Tier 3 demand, which is in line with empirically observed demand after grid electrification in Rwanda (see Table S6 caption for details) to find that oversizing leads to a 128% increase in the LCOE for MG and a 121% increase for SAS respectively. These large cost increases underline the importance of reading cost estimates put forth in this article with the demand assumption in mind. If demand turns out to be substantially lower, costs increase accordingly.

We further observe that the LCOE is similar across tiers for the first 30% of the unelectrified population because these people live in urban or peri-urban areas with existing grid connections. Hence, least-cost electrification takes place via grid densification, which serves different tiers at similar cost. From the 30th to the 50th percentile, least-cost electrification still takes place via the main grid irrespective of the tier (see Fig. 2c), but grid extension is needed beyond densification. This results in a cost jump for Tier 2 electrification because low demand

households face a higher LCOE given the high upfront investment cost in grid extension. The cost-optimal electrification approach starts to differ between tiers at the median of the unelectrified population, where SAS are taking off for Tier 2 electrification, which in turn leads to the LCOE of Tier 2 electrification levelling off (see Fig. 2a). For Tier 2, SAS rapidly make up more than 80% of new connections after the median, whereas this development kicks in around the 75th percentile for Tier 3. MGs only play a major role in Tier 4 electrification, where MG and SAS supply roughly 90% of the electrification of the last decile.

Turning to the cost per person per day (pp/d), we generally estimate lower costs for lower tiers because demand is lower. The costs stay remarkably flat and low for Tier 2 electrification across all of SSA reaching only USD 5c pp/d or USD 18.25 pp over an entire year. For Tier 3 electrification, costs pp/d are starting to diverge substantially from Tier 2 around the 60th percentile with the final 25% of the population to electrify paying 2.3 times as much per person compared to Tier 2. There is a more substantial premium on Tier 4 electrification throughout the electrification curve, but the diversion from Tier 3 costs pp/d similarly starts around the 60th percentile. Costs then increase sharply and reach 7.8 times the cost of Tier 2 electrification for the final 25% of the population to electrify. These cost differences for high-cost areas point to the value of a differentiated electrification approach where high-cost areas have the option of lower tier electrification, especially because demand may turn out to be lower than anticipated[33–35].

While it is likely that SAS can contribute to keeping electrification costs low in rural and sparsely populated areas with weak institutional quality, we would like to stress three limitations of our approach. First,

as in any electrification model, we assume a specific household demand profile depending on the tier and the country. If people consume more or less than this profile, costs pp/d will change accordingly. Second, much of the SAS deployment is carried out by private companies. While these companies theoretically would be able to offer electricity at the cost shown in Fig. 2, the structure and competition of the market and regulatory oversight will determine potential additional margins to these costs. Third, our analysis assumes that governments build out the planned grid and finance this build out on public accounts, which results in lower costs of capital compared to the private sector. However, historically governments have struggled to carry out grid densification and expansion plans and, where implemented, less households than projected were reached with associated caveats on development outcomes[36,37]. Furthermore, utility investment in SSA is constrained by large debt burdens and poor cost recovery factors, which have worsened further due to Covid-19 for all but few utilities that heavily import fossil fuels, for which prices have declined substantially[38]. The costs reported in this study should therefore be read as a potential rather than a forecast of reality.

## Geographical variance

Having established LCOE and cost pp/d electrification curves, we next examine the between country variation in costs pp/d focusing on Tier 3 electrification. Figure 3a shows the average cost pp/d for least-cost electrification in 40 SSA countries excluding, Cape Verde, Comoros, Djibouti, Mali, Mauritius, Ivory Coast, Seychelles, Sao Tome & Principe, and Sierra Leonne (see Supplementary Table S7 for a tabular format). Note that these costs are averaged from spatially explicit modelling

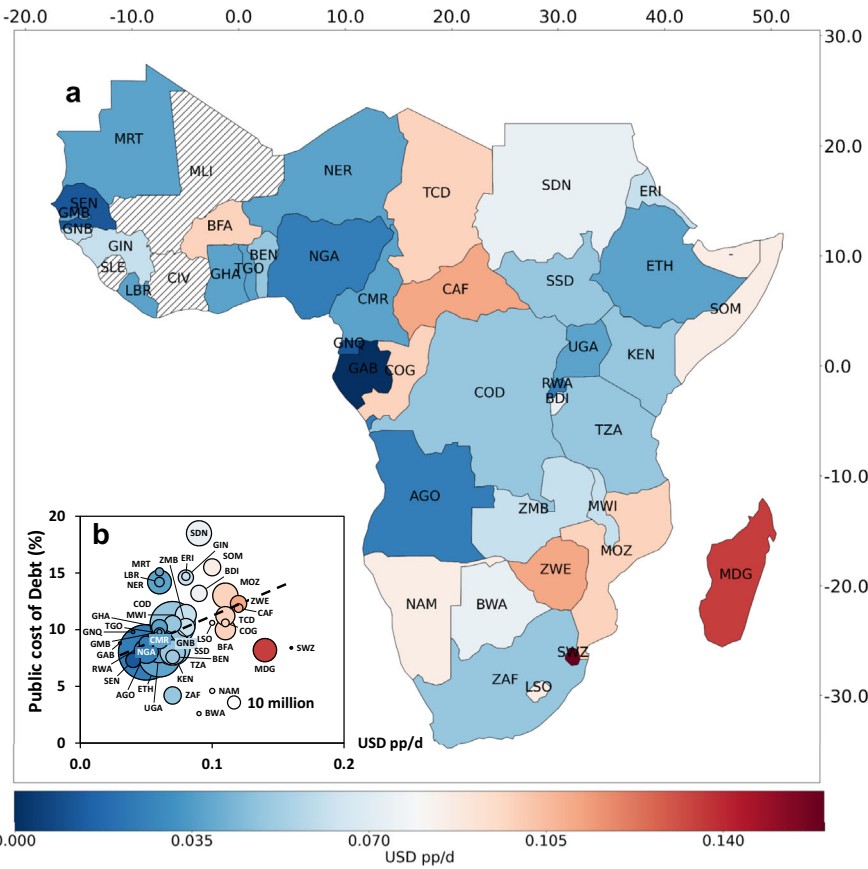

**Fig. 3 | Average cost of Tier 3 electrification per person per day by country.** **a** Average cost of the least-cost electrification approach to electrify everyone by 2030. **b** Average cost pp/d compared to the public cost of debt by country, a proxy for the institutional quality of the country. The size of bubbles in the scatter plot represents the total population to be electrified by 2030. The dashed line represents a linear trendline. The x and y axis on the map show the longitude and latitudes.

output as described in the methods and shown in Fig. S4 in the Supplementary Information. Costs are typically low in population centres and along grid lines and they are higher in more remote areas (see Fig. S4). By country, costs vary by a factor of 5.3 from USD 3c pp/d in Gabon to USD 16c pp/d in Eswatini. Beyond Gabon, Equatorial Guinea and Senegal exhibit costs below USD 5c pp/d too, whereas eight countries exhibit costs above USD 10c pp/d (Burkina Faso, Central African Republic, Chad, Madagascar, Mozambique, Republic of Congo, Eswatini, Zimbabwe). Figure 3b plots countries' average costs in USD pp/d versus the public cost of debt, a common indicator of institutional quality, which is also used as an input in the least-cost calculation to represent the investment environment[17]. We observe a strong positive correlation between the average cost pp/d for electrification and the public cost of debt, a proxy for the institutional quality of a country.

However, for some countries, such as Sudan (SDN), costs are relatively low despite a very high public cost of debt. 60% of new connections in Sudan are through SAS and households are large (6.5 people on average). These households use large SAS with lower investment cost per kW to satisfy demand, which lowers the LCOE. In addition, the high irradiation helps lower SAS LCOEs in Sudan. Conversely, Eswatini (SWZ) and Madagascar (MDG) the countries with the highest cost pp/d, do not face very high public costs of debt. In Eswatini, about 84% of new connections occur via the grid and these costs are distributed to a small number of people due to very low population density and small households. In Madagascar, the population density is higher, but households are small too (4.4 people on average), which again means that the costs are distributed to a smaller number of people (especially for grid extension). See Fig. S1 in the Supplementary Information for the full comparison of inputs by country.

Figure 3b also shows that a large part of the population to be electrified is concentrated in few countries. 475 million or 50% of population to be electrified by 2030 live in just five countries: the DRC (105 million), Ethiopia (94 million), Nigeria (158 million), Tanzania (64 million), and Uganda (54 million). These countries generally feature low average costs for Tier 3 least-cost electrification between five and seven cents pp/d (DRC 7c, Ethiopia 6c, Nigeria 5c, Tanzania 7c, and Uganda 6c, see Fig. 3a and Fig. S2 in the Supplementary Information). This range corresponds to annual costs between USD 18 and USD 26, which indicates that there may be a chance for electrification to proceed at relatively low cost per household[29].

Notwithstanding a low average cost pp/d for SSA and the countries with large populations to be electrified, there is substantial within country cost variation. Figure 4 shows the cost curve for each country in alphabetical order. While the cost curve for SSA as a whole resembles a flat S-curve (see Fig. 2b), we identify four types of cost curves on the country level (see Table 1 for a discussion). Across all countries, rural areas tend to face higher electrification costs (see Fig. S3 in the Supplementary Information).

First, twenty countries (full list, see Table 1; combined 565 million people to be electrified) exhibit flat cost curves in the beginning, which increase towards the tail end. This includes the five countries with the largest populations to electrify: Ethiopia, Nigeria, Tanzania and Uganda all show very flat cost curves for the first 40% of the population to be electrified, pointing to vast low-cost electrification opportunities in densely populated areas close to the grid via grid densification and extension. The DRC shows flat and low costs only for the first 20% of the population to be electrified, but costs remain relatively flat for the next 20% too with a more pronounced increase afterwards compared to the other countries. This is because the DRC has a low electrified population (19% in 2018), offering little opportunity for grid densification and extension. Second, eight countries (203 million people) feature cost curves that constantly increase throughout the population to be electrified (see e.g., Burundi). Third, six countries (122 million

people) show cost curves that increase in the beginning and flatten out towards the tail end (see e.g., Chad). Fourth, five countries (62 million people) show a stepwise cost curve whereas cost are flat in the beginning, increase sharply around the median of the unelectrified population (between 30th and 60th percentile) and flatten out again afterwards (see e.g., Namibia). We elaborate on these differences and their implications for policymakers in detail next.

## Discussion

To assess the relative cost of electrification, policymakers often rely on techno-economic least-cost electrification models[39,40]. In this analysis we have demonstrated that such models can provide a more complete picture of the challenge to reach SDG7 by focusing on key enabling factors beyond the least-cost choice of the electrification approach. First, such models can be used to consider the ability and willingness to pay for the electricity by the population to be electrified. Second, disaggregated analysis of modelling outcomes can help reveal between and within country heterogeneity in the electrification cost to consumers, which we find to be large. Third, such models can contrast different electrification pathways depending on tiers, thereby opening new opportunities to reach SDG7, such that consumers more likely will be able to pay for the electricity delivered.

We therefore propose using electrification cost curves for LCOE and electricity costs per person per day as a key metric for electrification planning. Establishing these cost curves, we find that the advent of low-cost off-grid electrification, such as standalone systems, has dramatically lowered the cost to electrify people living in remote areas with difficult institutional contexts (i.e., high cost of capital). We further show that the countries with the largest share of people without access to electricity in SSA exhibit lower than average costs pp/d for electrification. On average, we find that least-cost electrification in SSA costs 0.14 USD/kWh or 0.07 USD pp/d (0.20 USD/kWh or 0.03 USD pp/d for Tier 2 and 0.11 USD/kWh or 0.15 USD pp/d for Tier 4, see Supplementary Table S7). These cost estimates are sensitive to demand projections and if demand turns out to be substantially lower than expected, cost increases due to misguided planning and system oversizing can be large. The uncertainties around these cost estimates further increase for higher-cost areas (see Supplementary Fig. S6), which points to an additional challenge in achieving "the last mile" in electrifying SSA. Finally, we find that countries show very differently shaped cost curves in USD pp/d, which are outlined in Table 1.

Table 1 exemplifies policies that are conceivable to assist electrification. These need to be tailored to country specificities based on the electrification cost curves and local abilities and willingness to pay for electricity – and so does international policy support. Hence, just as any energy strategy, electrification planning needs to be spatially disaggregated to reflect variance between and within countries[5,21]. Optimising on LCOE only may lead to an inability to pay for consumers and/ or unsustainably high public subsidy costs. Reducing demand tiers provides an option for policymakers to keep cost within check, especially for countries where cost curves increase sharply. This path would lead to an even more prominent role for SAS in electrification of SSA (see Fig. 2c). Naturally, lower demand tiers imply lower levels of energy services, so policymakers in the respective countries will have to make decisions taking the trade-off between cost and energy service level into account, depending on their electricity access priorities and timelines.

Achieving 100% electrification in SSA at feasible cost demands executing grid extension plans swiftly using the ability of governments to raise money in capital markets at favourable rates. Where government budgets are in distress due to the economic downturn of Covid-19, international assistance should be provided for capital raising plans to densify and extend electricity grids and to improve utility performance, which many SSA governments grapple with due to dwindling

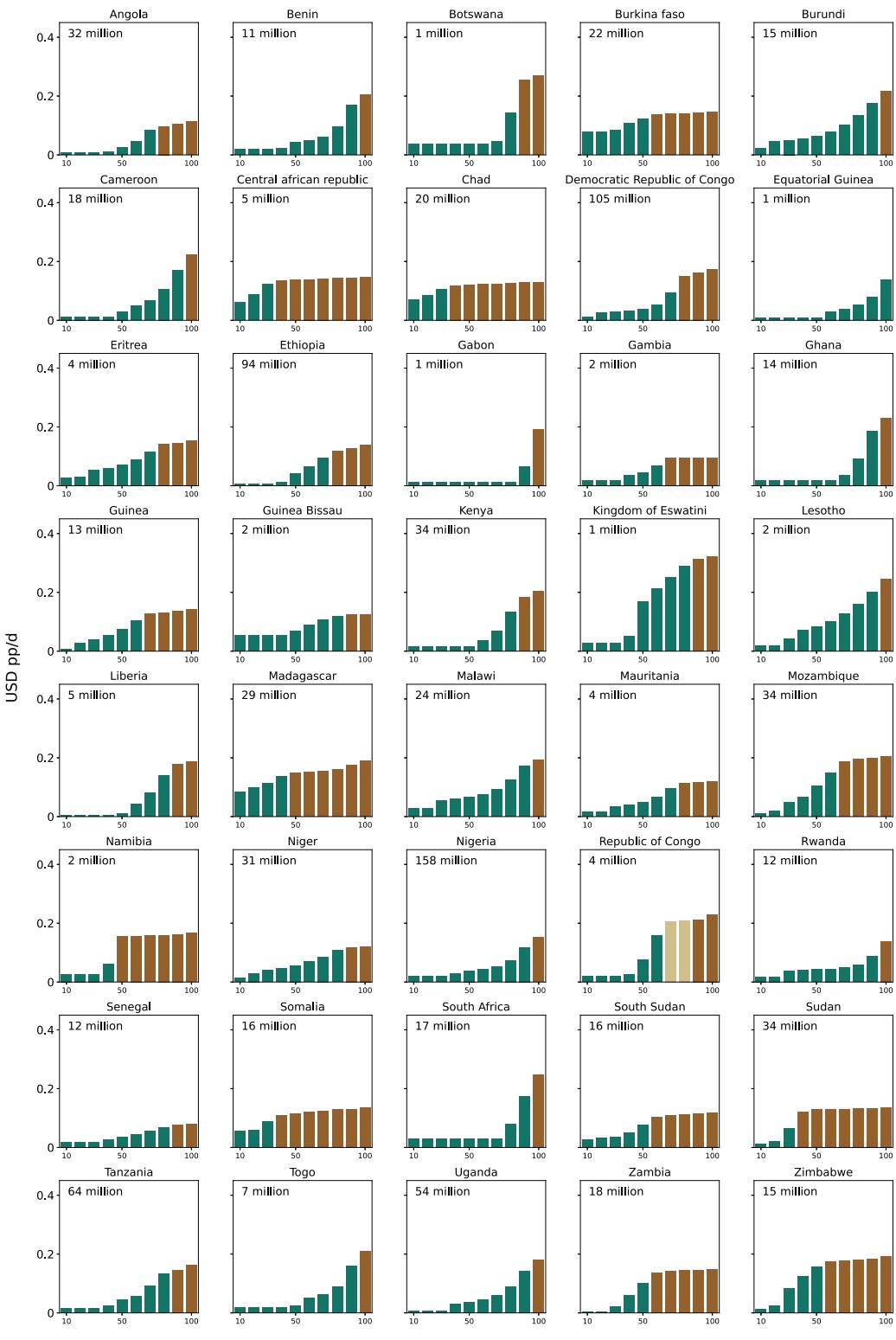

**Fig. 4 | Electrification cost curve (Tier 3) by country.** Costs are in USD pp/d, each bar represents a decile of the population to be electrified by 2030 to reach 100% electrification. The absolute of that population is provided in the top left of each diagram. Colours represent different electrification approaches and are assigned based on the dominant technology per decile.

revenues[38]. In turn, realising the potential of off-grid solutions depends on the ability of private companies to deploy SAS and MG at scale. Especially, SAS companies have managed to scale fast in the past, for example increasing their customer base 25-fold from 2015 to 2019 in Ghana, Kenya and Nigeria[4].

Continued fast deployment will only be possible if repayment rates are sufficient such that these private companies stay afloat without needing to increase the pricing. While existing data suggest very high repayment rates for solar lamps[41], these may become an issue for MG[42] and more generally as off-grid companies start serving more

**Table 1 | Electrification cost curve descriptions**

| Shape of the curve | 1<br>Flat → increasing | 2<br>Constant increasing | 3<br>Increasing → flat | 4<br>Flat → increasing → flat |
|---|---|---|---|---|
| Population affected | 565 million | 203 million | 122 million | 62 million |
| List of countries (in alphabetic order) | Angola, Benin, Botswana, Cameroon, Equatorial Guinea, Ethiopia, Ghana, Guinea Bissau, Kenya, Kingdom of Eswatini, Liberia, Malawi, Mauritania, Nigeria, Rwanda, Senegal, South Africa, Tanzania, Togo, Uganda. | Burundi, DRC, Eritrea, Gabon, Lesotho, Madagascar, Niger, Somalia. | Central African Republic, Chad, Guinea, Mozambique, South Sudan, Sudan. | Burkina Faso, Gambia, Namibia, Republic of Congo, Zambia, Zimbabwe |
| Policy implications (S = supply; D = demand, cf. ref. 21) | S1: Increase the efficiency and delivery of (public) utilities for the first part of the cost curve, mainly consisting of grid densification.<br>S2: Differentiated support schemes for (private) off-grid companies by region, tailored to cost levels for the second part of the cost curve.<br>D1: Differing demand subsidies for the highest-cost areas by country depending on the ability to pay. | S1: Regionally differentiated electrification plan with support schemes adjusted to the service provider (public or private) and the cost level throughout the cost curve.<br>D1: See column 1. | S1: See column 1.<br>S2: National support scheme for (private) off-grid companies for the second part of the cost curve (constant cost).<br>D1: National demand subsidies if off-grid cost levels remain too high for the population despite S2. | S1: See column 1.<br>S2: See column 3.<br>D1: See column 3. |

Ordered by affected population.

remote and less affluent areas. In such contexts, business model innovation, e.g., improving pay-as-you-go services for SAS, can reduce collection costs and increase sanction possibilities, which may contribute to realising sufficiently high repayment rates in the future. However, there remains a danger of underestimating the cost of electrification because such costs vary at the subnational level. One possible way of operationalizing this in future research could be via a cost of capital specific to regions as has been done in ref. 18. The extent to which moving to more remote and less affluent areas will lead to higher operation costs also depends on whether government electrification plans provide the necessary enabling environment, complemented by tailored international assistance. Such national and international policies can help lowering the cost of capital with associated cost savings as shown in Supplementary Fig. S5.

Additionally, more prevalent small-scale productive uses (e.g., from micro-businesses) could increase demand for electricity, which would reduce LCOEs, as shown in Fig. 2a. The extent of this effect is hard to quantify because predicting productive use spatially explicit for SSA is difficult and requires many assumptions. Furthermore, anchor loads, i.e., larger scale productive use applications, such as telecommunication towers, irrigation, or health care services, typically do not pay household tariffs for electricity. Rather, these service providers have tailored offtake agreements, or they even invest in electricity generation assets themselves. While, based on a need for policymaking[43], research has started developing tools to estimate future productive use locally[44] more research integrating different types of productive use into more macro-oriented models is required[45,46].

Finally, it is worth emphasising that the country differences in electrification costs also stem from differences in the institutional quality, reflected in different costs of capital in our analysis (see Fig. 3b). The affordability challenge of SDG7 becomes even more pressing in this context because the people facing the highest electrification costs are likely to live below the poverty line. More generally, policies that improve political stability, the rule of law, and institutional quality are likely conducive to reaching SDG7 and associated SDGs faster and at lower cost.

## Methods
### Electrification approaches
We model 100% electrification pathways for 40 sub-Saharan African countries. Electrification takes place in 2030 through grid extension (GE), mini-grids (MG) and standalone systems (SAS). GE includes densification, that is the main grid is extended to households without access to electricity in areas (clusters) that are already partly electrified, and extension, that is the main grid is extended to clusters that were previously unelectrified. Grid extension is limited to areas that are less than 50 km away from the existing or the planned grid. We modelled a pathway (extended area pathway, see ref. 17), where the planned grid is constructed in addition to the already existing grid infrastructure as per the World Bank's existing and planned grid infrastructure dataset[47]. MG encompasses solar PV plus battery powered MGs and hydro-powered MGs. The per kW cost for MGs is based on a single size (ca. 100 kW system) and the model does not distinctly size for different battery capacities, which is an important sizing parameter for the LCOE calculation. However, the distribution lines for MGs are sized depending on the population density within a cluster and the number of people per household. Overall, because a typical MG deployed in sub-Saharan Africa is smaller than 100 kW[48], the cost estimate is likely bullish. Results show a small role for MGs electrification despite this optimistic cost estimate. SAS includes solar PV plus battery powered standalone systems. The sizing of SAS for households depends on the solar irradiation and the number of people per household and costs per kW vary accordingly. We omit diesel powered electrification technologies mainly because newly deployed

private sector financed off-grid technologies are almost exclusively powered by renewable energy[17]. Table S2 and Table S8 in the Supplementary Information summarises the technical assumptions per electrification approach.

## Model setup

We used the modified[17] open-source integrated household geospatial electrification model OnSSET v.1.0 from the Global Electrification Platform. The key modification introduced in ref. 17 concerns the implementation of realistic financing conditions (i.e., cost of capital) that represent country and electrification approach specific risks (see Supplementary Table S8). Namely, we assume that GE is financed by the public sector, whereas off-grid electrification, MG and SAS, is financed by the private sector (i.e., "niche" scenario in ref. 17). This means that the cost of capital is lowest for grid electrification followed by SAS and MG because SAS companies can attract more debt (on average 50% compared to 0% for MG companies)[17]. The differences in cost of capital can also be interpreted as differences institutional risks. For example, fee collection is a challenge for off-grid electrification, which is exacerbated in countries with low institutional quality (conversely, high cost of capital). Similarly, fee collection is more challenging if many customers share the asset as in the case of MGs[42].

OnSSET has become a ubiquitous geospatial electrification modelling tool for academia[17,49–51] and international organisations working in electricity access in developing countries, such as the Global Electrification Platform by World Bank's Energy Sector Management Assistance Program (ESMAP). OnSSET calculates the least-cost electrification approach by area (referred to as a cluster) for a user specified target year – 2030 in the case of this study. The clusters are composed of cells which are estimated using population raster data at a granularity of 100 m × 100 m and include population growth (medium rate as per OnSSET) by country. The model then selects the least-cost electrification approach (i.e., GE, MG or SAS) for each cluster.

The levelized cost of electricity (LCOE) is given by Eq. 1.

$$LCOE = \frac{\sum_{m=1}^{n} \frac{I_m + O\&M_m - S_m}{(1+r)^m}}{\sum_{m=1}^{n} \frac{E_m}{(1+r)^m}} \tag{1}$$

$I_m$ is the investment cost for an electrification approach in year m, $O\&M_m$ is the operation and maintenance costs, $S_m$ is the salvage value (the value of the energy system at the end of its useful life), $E_m$ is the electricity generated, r the discount rate and n is the lifetime of the project in years. For the LCOE calculation, the GE LCOE is calculated by adding an estimated average grid electricity generation cost to the LCOE of transmitting and distributing electricity. The LCOEs for the off-grid electrification approaches are based on generation system costs, distribution infrastructure costs (MGs) and O&M costs. We model three different demand tiers (Tier 2, Tier 3, and Tier 4) as per the World Bank's Multi-tier framework (see Table S1 in the Supplementary Information for details)[22]. Demand is fixed per tier and country as per OnSSET and always remains within the MTF boundaries defined by the World Bank.

The excluded countries are not analysed because we do not have a complete country input dataset for this version of OnSSET v 1.0. Somalia and Somaliland are not split in our analysis because a cost of capital estimate is unavailable for Somaliland. Corresponding to SDG7, we set an electrification target of 100% in 2030.

## Capital expenditures

Defining future capital expenditures for each electrification approach is difficult because data is sparse. For GE, we assume constant grid capacity generation costs because despite large, planned capacity increases in some SSA countries, it is unlikely that the marginal cost of grid supplied electricity will change massively within the next few years. For MG and SAS, we follow a three-step approach. We use historical component cost data (i.e., modules, battery, inverter, etc.) from IRENA[52] and apply conservative cost reduction factors from UNDP[53] per electrification approach (see Table S3 in the Supplementary Information). We triangulate these results with the (few) other available sources on future MG and SAS cost for SSA in Table S4 in the Supplementary Information.

## Estimating LCOE curves

The model calculates a least-cost electrification for each cluster. We assembled cost curves by ordering clusters from smallest to largest LCOE across SSA. Using the number of people in each cluster, we then created population bins (deciles or ventiles depending on the plot). These bins are approximated and are not exact equal sizes due to computational limitations.

The weighted average LCOE (WLCOE in USD/kWh) is given by Eq. (2):

$$WLCOE_j = \frac{\sum_{i=1}^{n} LCOE_i \times energyperyear_i}{\sum_{i=1}^{n} energyperyear_i} \tag{2}$$

Where $i$ is the cluster ID and $n$ is the total number of clusters within a specified bin $j$. The country-level cost per person per day $C_{pp/d}$ (see Fig. 3) is calculated according to Eq. (3):

$$C_{pp/d} = \sum_{i=1}^{n} \frac{LCOE_i \times energyperyear_i}{newconnections_i \times 365} \tag{3}$$

Where $i$ represents a cluster and $n$ is the total number of clusters in a given country. New connections refer to the total number of newly electrified people per cluster and the energy per year is the energy consumed by these people in kWh.

We made further modifications to the OnSSET python code. Firstly, we update the SAS algorithm that estimates the system capacity per households. While the previous code estimated it by dividing the additional installed capacity within a specific cluster by the total cluster population (including already connected customers in 2018 and 2025), we divide the installed capacity by the new connections in 2030 only, in order to get an estimate of the costs for additional systems.

## Ethics and inclusion statement

This paper is part of a 3-year project on electricity access in sub-Saharan Africa funded by ETH for Development (ETH4D). The project is carried out by researchers at ETH Zurich, one of whom is affiliated with the Kigali Collaborative Research Centre in Rwanda (C.A.). The research is locally relevant for sub-Saharan African countries as it supports the advancement of electrification planning tools, which are often shaping electricity access policy in sub-Saharan Africa. Inputs into the analysis, particularly around the cost of capital assumptions, have been informed by extensive engagements with experts working in sub-Saharan Africa's electricity sector both locally and internationally.

The analysis in this paper relies on computational modelling and did not include or study participants at the individual level. It uses publicly available data by the World Bank, and empirical data informed by research that has been carried out by scholars in sub-Saharan Africa. As such, it does not capture local nuances, particularly around the households that would be consuming the electricity. For example, the model does not consider household-level load profiles, income, or individual preferences for electrification approaches. Moreover, the model also abstracts from the topography, which may influence electrification choices. Hence, any insights taken from this study should be contextualised carefully taking into account additional local evidence. Recommendations made based on the analysis offer a framework to

approach electrification challenges but are not prescriptive since different governments may have different electrification priorities. Part of the research has also been shared and discussed in workshops with policymakers in public administrations in different countries of sub-Saharan Africa. Finally, the analysis draws on insights from extensive local research in sub-Saharan African countries particularly around willingness and ability of customers to pay for electricity.

## Data availability
The country input data used in this study are available in the open access repository database which can be accessed on: https://www.research-collection.ethz.ch/handle/20.500.11850/527356. The additional cost input data are provided in the Supplementary Information.

## Code availability
The adapted OnSSET model used for this analysis can be accessed through the GitHub public repository (https://rb.gy/3ickh). This includes a short description of the modification compared to ref. 17, namely the accurate sizing of SAS for newly connected households.

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

## Acknowledgements

The authors would like to thank the team members of the Energy and Technology Policy Group and the Climate Finance Policy Group for feedback at internal seminars. This project has received funding from the Engineering for Development (E4D) Doctoral Scholarship Programme which is funded by the Sawiris Foundation for Social Development and the Swiss Agency for Development and Cooperation (C.A. and T.S.S).

## Author contributions

F.E., C.A, B.S and T.S.S developed the research idea and designed the research. C.A. ran the analysis, F.E., C.A., B.S and T.S.S. interpreted the results. F.E. wrote the manuscript with input and edits from C.A., B.S. and T.S.S.

## Competing interests

The authors declare no competing interests.
