## [Peer Review File · Nature Communications]

The Cost of Electrifying All Households in 40 sub-Saharan African Countries by 2030REVIEWER COMMENTS

Reviewer #1 (Remarks to the Author):

The paper provides cost estimates for reaching SDG 7 - universal access to electricity by 2030 - in sub-Saharan Africa in order to identify context-specific least cost electrification solutions. The topic is highly relevant since given today's investment commitments it is likely that SDG 7 will not be reached in SSA. The strength of the paper is the fine granularity of the estimates, providing least-cost curves for 40 different countries considering different electrification technologies and demand patterns. This enables decision-makers to develop context-specific solutions. The paper is clearly written and provides helpful figures.

I have two more substantial comments that I believe should be addressed before publication. Further smaller comments ensue.

1) The authors assume that all electricity supplied by a given electrification technology is effectively consumed by the customer. Yet, what we often see in rural SSA, are very low consumption levels even if the electricity source allows higher consumption levels (see for example Chaplin et al. 2017, Lenz et al. 2017, Peters and Sievert 2016, Taneja, J. 2018). Accounting for this, costs per kWh of high-capacity electricity sources (mainly on-grid, but also micro-grids) would be substantially higher than assumed in this study and the optimal electrification strategies might look differently. Would it be possible to show the sensitivity of results to different assumptions about actual consumption levels?

2) Another cost factor that I am not sure how it is accounted for is the level of management and operation costs. This is probably more complicated for decentralized solutions than for on-grid electrification. Fee collection in mini-grid settings for example is challenging (Peters, Sievert and Toman 2019) and probably also very context-dependent. Where do cost estimates for management and operation costs come from? Also, the authors mention "generally high repayment rates for off-grid electricity sources". They refer to a study that analyses repayment performance for pay-as-you-go solar lamps. Solar lamps are on the one hand the cheapest electrification sources and repayment of more substantial amounts might be very different. On the other hand, pay-as-you-go technology is the most advanced technological solution for addressing repayment. Do the authors account for this in their investment cost estimates and do comparable technologies exist for other off-grid technologies? I could imagine that the model substantially underestimates management and operation costs. Again, it would be interesting to see how sensitive the results are for different assumptions about these costs.

A very interesting result is that micro grids do not play any significant role in the scenario for Tier 3 electrification. Could you discuss a bit more where this result comes from? In the policy discourse mini-grids are often hailed for being an important intermediary electricity source for medium consumption levels.

I am surprised to see the authors chose different financing modes for on-grid (public finance) and off-grid (private investors). Even if this properly describes the status quo, it could in theory be different and might bias the analysis. How sensitive are results to this decision?

As the authors rightly point out, ability to pay for electricity still needs to be assessed and would be a logical next step with the study's results at hand. But at the same time, the paper makes statements like "electrification [...] may be possible without large public subsidies". I do not think this is correct. In many remote areas in SSA the population is so poor that they are not able (or willing) to pay even very simple solar lamps (see for example (Grimm et al. 2020)). Even Lighting Global, the World Bank Group's initiative to support off-grid solar products has started to acknowledge that subsidies are needed to achieve universal access. <https://www.lightingglobal.org/activities/pro-poor-activities/>

A sentence I don't understand (line 150): "This means that if governments and the private sector

manage to mobilize the initial investment requirement to electrify SSA, many consumers will likely be able to pay for Tier 2 electrification". Tier 2 electrification is most likely provided by Solar Home Systems – since these systems do have very low running costs it is obvious that the initial investment is the main challenge. I do not understand where this conclusion comes from and how it is related to the flat cost curve.

Chaplin, D., A. Mamun, A. Protik, J. Schurrer, D. Vohra, K. Bos, H. Burak, L. Meyer, A. Dumitrescu, C. Ksoll, and T. Cook (2017). Grid Electricity Expansion in Tanzania by MCC: Findings from a Rigorous Impact Evaluation. Report submitted to the Millennium Challenge Corporation, Washington, DC: Mathematica Policy Research.

Grimm, Michael, Luciane Lenz, Jörg Peters, and Maximiliane Sievert. 2020. 'Demand for Off-Grid Solar Electricity: Experimental Evidence from Rwanda'. *Journal of the Association of Environmental and Resource Economists* 7(3): 417–54.

Lenz, L., A. Munyehirwe, J. Peters and M. Sievert (2017). Does Large Scale Infrastructure Investment Alleviate Poverty? Impacts of Rwanda's Electricity Access Roll-Out Program. *World Development*, 89 (17): 88-110.

Peters, Jörg, and Maximiliane Sievert. 2016. 'Impacts of Rural Electrification Revisited – the African Context'. *Journal of Development Effectiveness* 8(3): 327–45.

Peters, Jörg, Maximiliane Sievert, and Michael A. Toman. 2019. 'Rural Electrification through Mini-Grids: Challenges Ahead'. *Energy Policy* 132: 27–31.

Taneja, J. (2018). If You Build It, Will They Consume? Key Challenges for Universal, Reliable, and Low-Cost Electricity Delivery in Kenya (No. 491).

Reviewer #2 (Remarks to the Author):

The manuscript "The Cost of Electrifying sub-Saharan Africa by 2030" carries out a sub-Saharan Africa-wide spatial modelling of electricity access with the OnSSET model (Mentis et al. 2017).

Novelty: the paper presents some element of novelty, such as the calculation of cost curves for electricity access and interesting post-model run analysis, such as the visualisation of cost-optimal approaches. However, several studies have already carried out SSA-wide modelling of electricity access using a bottom-up approach.

- Dagnachew et al. <https://www.sciencedirect.com/science/article/pii/S0360544217313282>

- Szabò et al. <https://iopscience.iop.org/article/10.1088/1748-9326/6/3/034002/meta>

- Pappis. <https://iopscience.iop.org/article/10.1088/2634-4505/ac7900/meta>

- Falchetta et al. <https://www.sciencedirect.com/science/article/pii/S097308262100048X>

I am very surprised that the authors do not refer to and discuss most this literature in the introduction to at least identify the key novelties of their approach.

Introduction [lines 55-63]: while I agree that the "affordability" gap is an issue, equally important challenges at stake are (1) scarce financing flows (private companies, PPA, PPP) and capital attractiveness of national investment environments; (2) business models to overcome uptake entry barriers of the poor; (3) deficit of public utilities with limited investment capacity. I recommend the author to discuss these crucial aspect with sufficient reference to the existing literature on these

topics.

Methods: the analysis only focuses on scenarios of demand from the residential sector. Yet, a number of studies have highlighted the crucial importance of productive uses of energy and other sectors (e.g. see <https://iopscience.iop.org/article/10.1088/1748-9326/ac0cab/meta>, <https://iopscience.iop.org/article/10.1088/2634-4505/ac611e/meta>, <https://iopscience.iop.org/article/10.1088/2634-4505/ac5fb2/meta>, <https://www.sciencedirect.com/science/article/pii/S0973082622001831>) for generating a realistic estimate of the latent energy demand. Neglecting demand from those sectors leads to (i) underestimations of total electricity demand; (ii) misleading LCOE maps; (iii) incomplete investment requirements assessments.

Therefore, I recommend the authors to use existing approaches (or find a novel methodology) to estimate local non-residential latent demand and include at least a scenario considering it to understand its relevance on top of residential demand only.

Results:

- Can you show maps of the sub-national optimal electrification technologies and LCOE variability over space? Previous research shows that there is huge variation within e.g. urban and remote rural areas within the same country.

-2030 is coming closer and closer, and it is clear (e.g.

https://www.afdb.org/fileadmin/uploads/afdb/Documents/Publications/Missing_the_Mark_Gaps_and_Lags_in_Disbursement_of_Development_Finance_for_Energy_Access.pdf,

<https://www.sciencedirect.com/science/article/pii/S25903322203014339>) that in several countries progress is too slow and unequal to meet universal electricity by 2030: why not also consider a more flexible horizon to see where we are heading to if investment does not ramp up?

- Sensitivity analysis: please consider carrying out sensitivity scenarios to evaluate the impact of cost parameters and other crucial techno-economic parameters, including the discount rate

Altogether, I believe that the manuscript cannot be published in the current form as it lacks both in sufficient novelty and depth of the analysis. I would be happy to reconsider a substantially revised version of the paper addressing the comments here above.

Reviewer 1

Comment	Response
The paper provides cost estimates for reaching SDG 7 - universal access to electricity by 2030 - in sub-Saharan Africa in order to identify context-specific least cost electrification solutions. The topic is highly relevant since given today's investment commitments it is likely that SDG 7 will not be reached in SSA. The strength of the paper is the fine granularity of the estimates, providing least-cost curves for 40 different countries considering different electrification technologies and demand patterns. This enables decision-makers to develop context-specific solutions. The paper is clearly written and provides helpful figures. I have two more substantial comments that I believe should be addressed before publication. Further smaller comments ensue.	We thank the reviewer for the positive and encouraging overall assessment and we hope to address all remaining concerns with the detailed point-by-point responses provided below. Changes in the revised manuscript are highlighted in yellow.
1) The authors assume that all electricity supplied by a given electrification technology is effectively consumed by the customer. Yet, what we often see in rural SSA, are very low consumption levels even if the electricity source allows higher consumption levels (see for example Chaplin et al. 2017, Lenz et al. 2017, Peters and Sievert 2016, Taneja, J. 2018). Accounting for this, costs per kWh of high-capacity electricity sources (mainly on-grid, but also micro-grids) would be substantially higher than assumed in this study and the optimal electrification strategies might look differently. Would it be possible to show the sensitivity of results to different assumptions about actual consumption levels?	We thank the reviewer for this remark regarding electricity consumption. Effectively, the uncertainty around future consumption is large with some scholars arguing for the incorporation of higher consumption levels, including productive use (see e.g., Reviewer 2), while others (mostly the empiricists) argue that consumption levels may turn out lower than anticipated. We generally agree with the reviewer that there is more evidence for the latter, particularly in ex-post evaluations of electrification programs. In fact, one of the key contributions “electrification cost curves” can bring to the policy debate is a differentiated take on consumption by making the cost differences along the unelectrified population visible. Essentially, the calculation of the cost curve for different demand tiers is a sensitivity of the results to different consumption levels (see Supplementary Table S1 for consumption levels and Table S7 for resulting LCOEs and costs pp/d). Required investments to reach 100% electrification by 2030 are \$87bn for Tier 2, \$203bn for Tier 3, and \$408bn for Tier 4. We added these figures to the revised manuscript (line 142-145).

As the reviewer points out, costs (i.e., LCOE) are substantially higher for lower tiers. Consequently, the optimal electrification strategy includes more SAS in lower tiers (see Fig. 2) and costs pp/d are lower (because consumption levels are lower), see e.g., line 192: “The costs stay remarkably flat and low for Tier 2 electrification across all of SSA reaching only USD 5c pp/d or USD 18.25 pp over an entire year.”

We acknowledge that changing tiers “only” accounts for a situation where the planning is adjusted to lower expected demand. However, in reality, planning may be off because demand was expected to be higher than it turns out to be. In such a situation, two factors cause costs. First, the suboptimal electrification approach may be chosen (e.g., MG instead of SAS). Second, even if the approach is right, the system may be oversized. Both factors increase LCOE, which we demonstrate in additional analyses discussed in a new results paragraph (line 151-168), shown in Supplementary Tables S5 and S6 and mentioned in the revised abstract and discussion. Namely, we show that choosing the electrification approach assuming Tier 3 demand leads to a 32% increase in the LCOE if demand is only Tier 2 (86% for Tier 4). Moreover, we use a representative cluster to show the cost increase if households show lower consumption levels than expected. We model this by halving Tier 3 demand but sizing the systems to meet Tier 3 demand, which is in line with empirically observed demand from grid connected households, thus potentially still an upper bound. LCOEs increase by a factor of 2.3 for MG and 2.2 for SAS respectively.

Moreover, we have added the suggested citations to solidify two points. First, we explicitly linked the issue of affordability to empirically observed electricity consumption citing Lenz et al. 2017 and Taneja 2018 (line 151-152 linked to previous paragraph). We also cite these additional references in the discussion of costs pp/d (line 201-202). Second, we stress that assuming the government builds the planned grid may be overly optimistic (line 210-217) citing Chaplin et al. 2017 and Peters and Sievert 2016.

	Additionally, it is important to note that the results shown in Figure 2 reflect the least cost electrification approach. The point highlighted by the reviewer about expected high costs if grid were used to electrify customers is observed in Figure 1 where we show the LCOEs assuming different electrification approach mixes. For example, electrifying the last 300 million people via grid would cost roughly 0.38 – 100 \$/kWh, adding MG reduces this to 0.25 - 7 \$/kWh and adding SAS further reduces the cost to 0.18 – 0.50 \$/kWh. Finally, we mention the fact that each result is based on an assumed demand tier, i.e., consumption level (line 145-150) and provide tier descriptions in Table S1. Kindly note that we do not include Tier 1, which is essentially solar lamps, and Tier 5, which are very large appliances with capacities greater than 2 kW providing 23 hours of electricity. These often include productive use, whereas we aim at modelling household electrification that can be reached realistically by 2030. We decided to make this point clearer by adding households to the revised title (see also response to comment #3 from reviewer 2).
2) Another cost factor that I am not sure how it is accounted for is the level of management and operation costs. This is probably more complicated for decentralized solutions than for on-grid electrification. Fee collection in mini-grid settings for example is challenging (Peters, Sievert and Toman 2019) and probably also very context-dependent. Where do cost estimates for management and operation costs come from? Also, the authors mention “generally high repayment rates for off-grid electricity sources”. They refer to a study that analyses repayment performance for pay-as-you-go solar lamps. Solar lamps are on the one hand the cheapest electrification sources and repayment of more substantial amounts might be very different. On the other hand, pay-as-you-go technology is the most advanced technological solution for addressing repayment. Do the authors account for this in their investment cost estimates and do comparable technologies exist for other off-grid technologies? I could imagine that the model substantially underestimates management and operation costs. Again, it would be interesting to see how sensitive the	We thank the reviewer for these very interesting and relevant points. Management and operation costs, including the cost to collect and risk of not collecting payment, are represented in two model parameters. First, the model uses percentages of CAPEX as O&M assumptions. These are 2% for SAS, 1.5% for MG CAPEX and 2% for MG T&D, and 2% for grid T&D (we added a column in Supplementary Table S2 for O&M assumptions). For the grid generation capacity, the model uses an estimated country specific grid LCOE provided by the World Bank, which represents the total cost of the existing generation stock in generating electricity (see https://electrifynow.energydata.info). As such, future cost savings are considered to the extent that O&M costs decline in proportion with CAPEX costs. If O&M costs were to decline faster as e.g. pay-as-you-go technologies become cheaper, we could slightly overestimate cost (and vice versa). However,

results are for different assumptions about these costs.	because O&M accounts for a small fraction of total cost, any changes in O&M costs have only a very small impact on our results. Second, repayment risk is accounted for in the cost of capital (CoC) that varies by country and electrification approach according to Agutu et al. 2022 (ref. 17). The country variation represents differences in institutional quality, resulting in different investment risks. Fee collection risk is much higher in countries with high CoC because typically household incomes are lower, because the infrastructure will be less developed (e.g., access to remote areas) and because recourse possibilities in case of non-compliance are sparse (cf. what is referred to as “end-user credit risk” in ref. 52 (UNDP 2019)). We also vary the CoC by electrification approach with MG facing a higher CoC because MG companies cannot attract the debt levels that SAS companies can (cf. methods in Agutu et al. 2022). Hence, we believe that this variation in the CoC contributes to representing fee collection risks and difficulties more accurately compared to the conventional models that use a uniform CoC. We mention these points in the revised methods (line 403-407) and refer to these lines in the discussion, also making it clear that the high repayment rates refer to solar lamps only (line 340-353). Moreover, we added two sentences suggesting future research to capture such subnational variance, which we believe is beyond the scope of a paper covering all of SSA. Finally, we show a sensitivity analysis in Supplementary Figure S5. It shows that the CoC of SAS has a particularly large impact on overall LCOE for electrifying SSA. This underlines the importance of capturing the challenges for off-grid electrification expansion accurately. However, the exact costs of such expansions are impossible to quantify because they are highly context dependent.
A very interesting result is that micro grids do not play any significant role in the scenario for Tier 3 electrification. Could you discuss a bit more where this result comes from? In the policy discourse mini-grids are often hailed for being an important intermediary electricity source for medium consumption levels.	We thank the reviewer for this remark. Indeed, mini grids are often proposed as a cost-effective means of electrification in the policy discourse. One of the key reasons for this is that current models do not differentiate the cost of capital between different electrification approaches (compare our above response). Yet, data shows

	that it is very difficult to attract commercial capital for mini-grids and the financing conditions are considerably worse compared to SAS. This is discussed in detail in Agutu et al. 2022 (ref. 17). We added a few sentences on this point in the result discussion on Figure 2 (line 145-150) but prefer to abstain from a more in-depth discussion because we believe the key contribution of this paper lies in the establishment of electrification cost curves and the associated policy implications.
I am surprised to see the authors chose different financing modes for on-grid (public finance) and off-grid (private investors). Even if this properly describes the status quo, it could in theory be different and might bias the analysis. How sensitive are results to this decision?	We thank the reviewer for this comment. There is solid evidence that financing modes are different for different electrification approaches as described in Agutu et al. 2022 (ref. 17). Yet, policymakers may choose to de-risk certain electrification approaches, lowering the financing costs. Such scenarios are presented in Agutu et al. 2022, where the authors contrast a public financing scenario (public finances all electrification) to the status quo (termed “niche”) and a potential future scenario where off-grid financing becomes mainstream (i.e., lowering the financing cost because large risk averse investors start investing). The results in Agutu et al. 2022 demonstrate how electrification varies according to these scenarios. We provide additional sensitivity analyses in Supplementary Figures S5 and S6. Because the key contribution of this paper is the establishment of cost curves, we believe that the sensitivity analysis provided in Figure S6 is most useful. It clearly shows the robustness of our findings. This is discussed in the text on Figure 2 (line 108-123).
As the authors rightly point out, ability to pay for electricity still needs to be assessed and would be a logical next step with the study’s results at hand. But at the same time, the paper makes statements like “electrification [...] may be possible without large public subsidies”. I do not think this is correct. In many remote areas in SSA the population is so poor that they are not able (or willing) to pay even very simple solar lamps (see for example (Grimm et al. 2020)). Even Lighting Global, the World Bank Group’s initiative to support off-grid solar products has started to acknowledge that subsidies are needed to achieve universal access.	We thank the reviewer for this note of caution. We agree that the statement needed to be toned down and removed the reference to public subsidies: “This range corresponds to annual costs between USD 18 and USD 26, which indicates that there may be a chance for electrification to proceed at relatively low cost per household.” We also referenced the study by Grimm et al. 2020 in this instance and as an additional reference in the discussion on willingness to pay (line 254-255). We discuss the need for public support policies (domestic and international) extensively in the discussion and by no means did we intend to suggest that

https://www.lightingglobal.org/activities/pro-poor-activities/	public support is no longer necessary for electrification in SSA.
A sentence I don't understand (line 150): "This means that if governments and the private sector manage to mobilize the initial investment requirement to electrify SSA, many consumers will likely be able to pay for Tier 2 electrification". Tier 2 electrification is most likely provided by Solar Home Systems – since these systems do have very low running costs it is obvious that the initial investment is the main challenge. I do not understand where this conclusion comes from and how it is related to the flat cost curve.	We thank the reviewer for this clarification questions. Indeed, the wording was unclear. The intent was to stress that electrification is not guaranteed simply because the cost per person per day seems affordable. We believe that this point is sufficiently clear from the rest of the manuscript and indeed it does not link to the discussion of cost curve results. We therefore removed the sentence.
Chaplin, D., A. Mamun, A. Protik, J. Schurrer, D. Vohra, K. Bos, H. Burak, L. Meyer, A. Dumitrescu, C. Ksoll, and T. Cook (2017). Grid Electricity Expansion in Tanzania by MCC: Findings from a Rigorous Impact Evaluation. Report submitted to the Millennium Challenge Corporation, Washington, DC: Mathematica Policy Research. Grimm, Michael, Luciane Lenz, Jörg Peters, and Maximiliane Sievert. 2020. 'Demand for Off-Grid Solar Electricity: Experimental Evidence from Rwanda'. Journal of the Association of Environmental and Resource Economists 7(3): 417–54. Lenz, L., A. Munyehirwe, J. Peters and M. Sievert (2017). Does Large Scale Infrastructure Investment Alleviate Poverty? Impacts of Rwanda's Electricity Access Roll-Out Program. World Development, 89 (17): 88-110. Peters, Jörg, and Maximiliane Sievert. 2016. 'Impacts of Rural Electrification Revisited – the African Context'. Journal of Development Effectiveness 8(3): 327–45. Peters, Jörg, Maximiliane Sievert, and Michael A. Toman. 2019. 'Rural Electrification through Mini-Grids: Challenges Ahead'. Energy Policy 132: 27–31. Taneja, J. (2018). If You Build It, Will They Consume? Key Challenges for Universal, Reliable, and Low-Cost Electricity Delivery in Kenya (No. 491).	We thank the reviewer for the excellent references, which are all included in the revised manuscript as described in the point-by-point responses above.

Reviewer 2

Comment	Response
Novelty: the paper presents some element of novelty, such as the calculation of cost curves for electricity access and interesting post-model run analysis, such as the visualiation of cost-optimal approaches. However, several studies have already carried out SSA-wide modelling of electricity access using a bottom-up approach. - Dagnachew et al. https://www.sciencedirect.com/science/article/pii/S0360544217313282 - Szabò et al. https://iopscience.iop.org/article/10.1088/1748-9326/6/3/034002/meta - Pappis. https://iopscience.iop.org/article/10.1088/2634-4505/ac7900/meta - Falchetta et al. https://www.sciencedirect.com/science/article/pii/S097308262100048X I am very surprised that the authors do not refer to and discuss most this literature in the introduction to at least identify the key novelties of their approach.	We thank the reviewer for the time devoted to the manuscript. We are encouraged by the positive overall assessment and thank her/him for the comments, which have improved the paper substantially. We believe that this paper makes three key contributions beyond the existing literature. First, it proposes electrification cost curves as a concept for research and policymaking to analyze options for reaching 100% electrification (as opposed to mostly maps in the existing literature showing cheapest options spatially for a given year). Second, it provides estimates on LCOE and the cost per person per day (the latter is often disregarded) for these cost curves. Importantly, we provide national cost curves, which is crucial for policymakers, while much research focuses on the overall picture. Third, we discuss the policy implications of these curves with an emphasis on pathways. While many papers look at the costs of reaching a 2030 endpoint, we believe that a crucial value in these cost curves lies in outlining policy options along the way more clearly. For example, the tier switch (line 324-326) is a policy option that is discussed. Nonetheless, we acknowledge the existing literature on electrification, and we have made sure to cite the references provided by the reviewer where appropriate. While some papers show demand tier differences (e.g., Dagnachew et al. 2017) or run demand scenarios (e.g., Pappis 2022), none of the papers propose or estimate electrification cost curves. Except for Falchetta et al. 2021, the papers do not differentiate discount rates, which we believe is crucial to produce realistic estimates (Falchetta et al. 2021 estimates subnational discount rates, which is very interesting and which we refer to in the revised discussion). We refrain from citing Szabo et al. 2011 because the focus of this paper lies in comparing diesel and PV options for off-grid electrification (no SAS), whereas we focus on low-emission electrification approaches.

Introduction [lines 55-63]: while I agree that the “affordability” gap is an issue, equally important challenges at stake are (1) scarce financing flows (private companies, PPA, PPP) and capital attractiveness of national investment environments; (2) business models to overcome uptake entry barriers of the poor; (3) deficit of public utilities with limited investment capacity. I recommend the author to discuss these crucial aspect with sufficient reference to the existing literature on these topics.	We thank the reviewer for this comment. We focus on the “affordability” gap in the introduction because this gap directly links to the analysis we provide (see also the points by Reviewer 1). Electrification cost curves can provide insights to policymakers regarding how to consider affordability more strategically. We took care to expand the discussion on the three aspects mentioned by the reviewer, including additional references. First, we added total investment needs according to tier in the revised manuscript and discuss the difficulty to attract such investments including additional references (line 142-145). We also mention the point in the discussion. Second, we mention business model innovation as a key driver very early in the manuscript including key references. We refer to this again in the discussion in the revised manuscript (line 343-346). Third, we explicitly discuss the high debt burden and poor cost recovery factors that most utilities in SSA face, and which have worsened due to Covid-19, as a caveat to our cost estimates (line 215-217). A recent World Bank report (2021) provides detailed information on this.
Methods: the analysis only focuses on scenarios of demand from the residential sector. Yet, a number of studies have highlighted the crucial importance of productive uses of energy and other sectors (e.g. see https://iopscience.iop.org/article/10.1088/1748-9326/ac0cab/meta, https://iopscience.iop.org/article/10.1088/2634-4505/ac611e/meta, https://iopscience.iop.org/article/10.1088/2634-4505/ac5fb2/meta, https://www.sciencedirect.com/science/article/pii/S0973082622001831) for generating a realistic estimate of the latent energy demand. Neglecting demand from those sectors leads to (i) underestimations of total electricity demand; (ii) misleading LCOE maps; (iii) incomplete investment requirements assessments. Therefore, I recommend the authors to use existing approaches (or find a novel methodology) to estimate local non-residential latent demand and include at least a scenario considering it to understand its relevance on top of residential demand only.	We thank the reviewer for this comment, which we address in three points. We focus on SDG7, particularly target 1, which is commonly referred to as SDG7: “By 2030, ensure universal access to affordable, reliable and modern energy services”. We added households to the title of the paper and clarified that we focus on target 1 in the introduction (line 39-41). This should make the focus of the paper and the analysis clearer. Besides the link to target 1 in SDG7, we focus on households for three main reasons. First, the evolution of productive use in SSA is very difficult to predict up to 2030. Particularly, in rural areas, it is challenging if not impossible to estimate what productive uses will be and where these will be needed to what extent. We would either need geographically explicit information on demand from productive use or we would need to use generalized assumptions, which we think will not provide additional information. Second, additional demand from productive use lowers the LCOE (see below) but it is unclear by how much. This is namely because productive use usually does not come

from households but from small businesses. However, such businesses do not pay the same tariff as households do (at least in most off-grid business models that we know of) and hence while demand may go up, the additional demand from productive use pays a lower tariff. This makes any statement on the effect on the LCOE highly speculative. For illustration, imagine the use case of a telecommunications tower. The operating company will either invest in electricity generation capacity itself, or it will secure larger procurement contracts with companies that provide off-grid electricity locally (e.g., through MG). Representing such arrangements is out of the scope of our effort in this paper but we refer to them in the revised manuscript (line 357-360). Third, it is unclear whether demand is underestimated due to future productive use or overestimated due to difficulties in the ability or the willingness to pay, which has often been observed empirically (please see first comment from Reviewer 1 and our response).

Having said that, we sense that underlying the reviewer's comment is a broader question around the sensitivity of our results to demand levels. We would like to emphasize that electrification cost curves across different tiers can be interpreted in view of productive use too. For example, a country's policymaker may want to look at electrification costs if a certain share of communities gain access to electricity levels that can serve productive use. As such, a policymaker may decide to plan for different electrification tiers in different geographic locations. While predicting the exact demand per location is difficult, research such as Falchetta et al. 2021 can help deriving heuristics as to where future demand could be more likely. As shown in Figure 3 of the above-mentioned paper, population density may be a good first approximation of productive use (regions around Nairobi and Kisumu). Such proxies could help providing more information to policymakers where higher tier electrification may be useful. We dedicated a new paragraph in the revised discussion to this point (line 354-364) and included all the suggested references, which were very useful pointers to the current state of the literature on latent demand estimation.

Results: - Can you show maps of the sub-national optimal electrification technologies and LCOE variability over space? Previous research shows that there is huge variation within e.g. urban and remote rural areas within the same country.	We thank the reviewer for this question. We model optimal electrification approach for clusters composed of cells with a granularity of 100m x 100m for all of SSA. The electrification cost curves are bunched together in ventiles (Figure 2) or deciles (Figure 4, by country) and represent population percentiles, which are constructed from the modelling by cluster. Hence, the subnational variance is visible in the cost curves shown in Figure 4. Furthermore, we explicitly show which deciles are predominantly rural versus urban in Figure S3 in the Supplementary Information. We added an additional Figure S4 in the Supplementary Information to the revised manuscript, where we show the modelling output for an exemplary country, Nigeria. Namely, we show how the cost per person per day is spatially explicit with lowest costs in population centers and along grid lines (visible as “arteries”) and higher costs in more remote areas.
-2030 is coming closer and closer, and it is clear (e.g. https://www.afdb.org/fileadmin/uploads/afdb/Documents/Publications/Missing_the_Mark_Gaps_and_Lags_in_Disbursement_of_Development_Finance_for_Energy_Access.pdf, https://www.sciencedirect.com/science/article/pii/S25903322203014339) that in several countries progress is too slow and unequal to meet universal electricity by 2030: why not also consider a more flexible horizon to see where we are heading to if investment does not ramp up?	We thank the reviewer for this comment. One of the motivations for this paper is indeed showing pathways rather than end points only. We believe that electrification cost curves provide crucial information to policymakers regarding how to get to 100% electrification. For the purpose of this paper and in accordance with SDG7, we fix 2030 as the year to achieve 100% electrification. However, one could set the target to a later year, which would lower costs as the cost of SAS and to some extent MG too is expected to decrease substantially with increasing deployment and consequential experience over time. Rather than varying the time horizon, we think that outlining different cost paths by tier provides policymakers with different paths to ramp up investment. For example, if electrification is to achieve Tier 2 only in the first instance, total investment is lower (line 143). Alternatively, electrification could start with higher tiers in lower cost regions and move to lower tiers in higher cost regions resulting in a total investment cost between the tiers (cf. policy option discussion, line 324-326). Finally, the last resort option if investment does not ramp up enough is to sacrifice 2030 as the target year to achieve 100% electrification by lowering the electrification target or increasing the time

	horizon. However, we prefer showing different pathways of getting to this target including policy options to facilitate these pathways (see Table 1).
- Sensitivity analysis: please consider carrying out sensitivity scenarios to evaluate the impact of cost parameters and other crucial techno-economic parameters, including the discount rate	We thank the reviewer for this remark on sensitivities. We added Figure S5 and Figure S6 to show the sensitivity of the electrification model to inputs. Figure S5 shows that SAS input parameters (CAPEX and cost of capital) have the highest impact on average LCOE. This is because high-cost regions are electrified with SAS, hence, these costs matter disproportionately for the overall cost of electrifying SSA. The good news is that the overall LCOE would only increase by 8% if the CAPEX of SAS were to be 20% higher than assumed (see Table S4 for cost inputs). In turn, average LCOE reductions of 20% are possible if CAPEX are 20% lower than anticipated and substantial cost savings are possible with de-risking investments (i.e., lowering the cost of capital) too. Grid and T&D infrastructure input costs have a relatively large impact too because a large share of the unelectrified population is projected to receive grid electrification. Finally, MG cost inputs matter proportionally less because fewer people are electrified with MG at the reference Tier 3. In accordance with the above, the sensitivities for the entire curves are very robust. Variance is higher for higher-cost regions (i.e., towards the right of electrification cost curves in Figure S6). This result is interesting as it shows that as electrification reaches higher-cost (i.e., typically more remote) areas, the uncertainties about costs also increase. We discuss this finding in an additional paragraph in the revised manuscript (line 108-123) and refer to it in the discussion too (line 313-315).
Altogether, I believe that the manuscript cannot be published in the current form as it lacks both in sufficient novelty and depth of the analysis. I would be happy to reconsider a substantially revised version of the paper addressing the comments here above.	We thank the reviewer for the critical comments, which have helped to improve the manuscript substantially. We hope to have addressed the comments comprehensively with the implemented revisions.

REVIEWER COMMENTS

Reviewer #1 (Remarks to the Author):

Thank you for the revision. My comments have been addressed satisfactorily.

Reviewer #2 (Remarks to the Author):

I very much appreciate the authors' effort to address the first round of review's comments, for instance by being more clear about the fact that this is a paper on HOUSEHOLD electrification and by carrying out sensitivity analysis.

I have one residual concern, which stems from the statements made in the following paragraph (page 14 of the paper, starting from line 357):

Furthermore, productive use applications, such as telecommunication towers, irrigation, or health care services, typically do not pay household tariffs for electricity. Rather, these service providers have tailored offtake agreements, or they even invest in electricity generation assets themselves. Still, based on a need for policymaking, research has started developing tools to estimate future productive use locally⁴³ because existing models rarely do⁴⁴. Yet, research also shows that including productive use only has a minimal impact on the economic viability of residential electrification⁴⁵, which may dampen hopes of productive uses drastically changing the picture on household electrification.

I disagree with most of the statement made here! Productive uses of electricity are in the first place activities that happen within or in the close surroundings of the household, and only at a second point evolve into formal businesses. For instance, small scale crop processing, barber shops, handcraft machinery, etc. Albeit small, these uses of energy can be defined 'productive' because they are targeted at guaranteeing the household an additional source of income. Perhaps what the authors are referring to is 'anchor loads' (e.g. healthcare facilities, telecom towers etc.), but this is rather different than productive uses of energy (see the literature, e.g.

<https://webassets.oxfamamerica.org/media/documents/Electrification-Morrissey-final.pdf>;

<https://www.sciencedirect.com/science/article/pii/S2667095X21000039>). Therefore, I believe the authors should revise their statement and their discussion of productive uses throughout the paper. Moreover, the authors cite a paper showing limited impact of productive uses for mini grids bankability, but what about other studies (e.g.

<https://www.sciencedirect.com/science/article/pii/S0973082620303409>) showing substantial role of productive uses for mini-grid operators' income?

Once the authors address this point thoroughly (also in relation to their analysis and results, rather than solely in terms of general discussion), I will be happy to recommend acceptance of the manuscript.

Comment	Response
Reviewer 1	
Thank you for the revision. My comments have been addressed satisfactorily.	We appreciate the reviewer’s constructive feedback, and we are glad that the reviewer was satisfied with the responses.
Reviewer 2	
I very much appreciate the authors’ effort to address the first round of review’s comments, for instance by being more clear about the fact that this is a paper on HOUSEHOLD electrification and by carrying out sensitivity analysis. I have one residual concern, which stems from the statements made in the following paragraph (page 14 of the paper, starting from line 357): Furthermore, productive use applications, such as telecommunication towers, irrigation, or health care services, typically do not pay household tariffs for electricity. Rather, these service providers have tailored offtake agreements, or they even invest in electricity generation assets themselves. Still, based on a need for policymaking, research has started developing tools to estimate future productive use locally⁴³ because existing models rarely do⁴⁴. Yet, research also shows that including productive use only has a minimal impact on the economic viability of residential electrification⁴⁵, which may dampen hopes of productive uses drastically changing the picture on household electrification. I disagree with most of the statement made here! Productive uses of electricity are in the first place activities that happen within or in the close surroundings of the household, and only at a second point evolve into formal businesses. For instance, small scale crop processing, barber shops, handcraft machinery, etc. Albeit small, these uses of energy can be defined ‘productive’ because they are targeted at guaranteeing the household an additional source of income. Perhaps what the authors are referring to is ‘anchor loads’ (e.g. healthcare facilities, telecom towers	We thank the reviewer for their critical remarks on productive use of energy (PUE). We agree that micro-businesses can play an important role. The criticized passage in our manuscript stems from a different definition of productive uses: While we were indeed mostly referring to larger scale (anchor) loads, the comment by Reviewer 2 focuses more on micro-businesses. We agree that this differentiation is important and took it into account in the revision (see fourth bullet of changes below). Importantly, our model focuses on households and as such, we argue that micro-businesses (mostly operating from home) are well represented by the different load tiers as defined by the World Bank. Our base case, Tier 3, includes smaller appliances such as a refrigerator (which is often used in micro-businesses). Tier 4 even includes heavier appliances like an iron or a sewing machine. In other words, we believe, that using Tier 3 as main specification and showing and discussing the effect of Tier 2 and 4 on the cost and technology choice prominently in Figure 2 addresses the reviewer’s concerns. Additionally, we would like to refer the reviewer to a recent pre-print publication specifically addressing the concern raised (https://www.researchsquare.com/article/rs-2888297/v1), the paper titled “Energy planning in Sub-Saharan African countries needs to explicit consider productive uses of electricity”, defines productive uses similarly to the definition provided by the reviewer i.e. “PUE is the additional electricity use of households on top of their consumption for end-use services in order to provide additional income through household enterprises. These activities include agricultural energy uses (such as water pumping for smallholder irrigation),

etc.), but this is rather different than productive uses of energy (see the literature, e.g.

<https://webassets.oxfamamerica.org/media/documents/Electrification-Morrissey-final.pdf>;

<https://www.sciencedirect.com/science/article/pii/S2667095X21000039>).

Therefore, I believe the authors should revise their statement and their discussion of productive uses throughout the paper. Moreover, the authors cite a paper showing limited impact of productive uses for mini grids bankability, but what about other studies (e.g.

<https://www.sciencedirect.com/science/article/pii/S0973082620303409>)

showing substantial role of productive uses for mini-grid operators' income?

Once the authors address this point thoroughly (also in relation to their analysis and results, rather than solely in terms of general discussion), I will be happy to recommend acceptance of the manuscript.

agro-processing (i.e. milling, husking, hulling), small-scale manufacturing (e.g. carpentry, tailoring, welding, looming) and service sector (like gastronomy, beauty salon, etc).” One key finding from the analysis is that the inclusion of PUE would increase the total household electricity demand in sub-Saharan Africa by 45%. Using this finding as the basis for analysis and considering that most rural household electricity demand is typically around Tier 2 (ca. 73 kWh/household/year), our inclusion of tier 3 and 4 more than accounts for productive uses.

To further address the reviewer's concerns, we changed the following:

- We explicitly highlight that variances in household electricity demand tiers can reflect variances in small-scale productive uses, including micro-businesses (lines 150 – 153).
- We have included a point on the potential benefit of productive uses due to increased demand during the day. Because this effect can be expected to play out for MG and SAS and it is difficult to quantify and generalize across SSA, we believe that the demand Tier variation remains a good representation of small-scale productive use. We also cite Hartvigsson et al. 2021 on this point as suggested (lines 153 – 156).
- We have revised our description on demand predictions highlighting that productive uses improve the economics of off-grid household electrification: “While demand can be underestimated if small-scale productive uses develop, empirically, it is often below the initial projections of household electrification programs in ex-post evaluations.” (lines 157 – 159).
- We have deleted the sentence on the minimal impact of productive uses on the economic viability of residential electrification and revised the paragraph to clearly delineate productive use from anchor loads (lines 361 – 370).

Finally, we agree that productive users can be a substantive part of the revenue/income source, however, income is not relevant for our paper as we focus on the cost side. The typically high risk of these small businesses does not

	increase bankability and therefore is unlikely to influence the cost of capital (see van Hove et al. 2022, cited in the previous version of the manuscript too).
--	--